# Early Permissiveness of Central Nervous System Cells to Measles Virus Infection Is Determined by Hyperfusogenicity and Interferon Pressure

**DOI:** 10.3390/v15010229

**Published:** 2023-01-13

**Authors:** Marion Ferren, Alexandre Lalande, Mathieu Iampietro, Lola Canus, Didier Decimo, Denis Gerlier, Matteo Porotto, Cyrille Mathieu

**Affiliations:** 1CIRI, Centre International de Recherche en Infectiologie, Team Neuro-Invasion, TROpism and VIRal Encephalitis, Université de Lyon, Inserm, U1111, CNRS, UMR5308, Université Claude Bernard Lyon 1, Ecole Normale Supérieure de Lyon, 69007 Lyon, France; 2CIRI, Centre International de Recherche en Infectiologie, Team Immunobiology of the Viral infections, Université de Lyon, Inserm, U1111, CNRS, UMR5308, Université Claude Bernard Lyon 1, Ecole Normale Supérieure de Lyon, 69007 Lyon, France; 3Center for Host-Pathogen Interaction, Columbia University Vagelos College of Physicians and Surgeons, New York, NY 10032, USA; 4Department of Pediatrics, Columbia University Vagelos College of Physicians and Surgeons, New York, NY 10032, USA; 5Department of Experimental Medicine, University of Campania “Luigi Vanvitelli,” 81100 Caserta, Italy

**Keywords:** measles virus, central nervous system infection, hyperfusogenicity, viral encephalitis, cell susceptibility, interferon treatment, organotypic culture

## Abstract

The cessation of measles virus (MeV) vaccination in more than 40 countries as a consequence of the COVID-19 pandemic is expected to significantly increase deaths due to measles. MeV can infect the central nervous system (CNS) and lead to lethal encephalitis. Substantial part of virus sequences recovered from patients’ brain were mutated in the matrix and/or the fusion protein (F). Mutations of the heptad repeat domain located in the C terminal (HRC) part of the F protein were often observed and were associated to hyperfusogenicity. These mutations promote brain invasion as a hallmark of neuroadaptation. Wild-type F allows entry into the brain, followed by limited spreading compared with the massive invasion observed for hyperfusogenic MeV. Taking advantage of our ex vivo models of hamster organotypic brain cultures, we investigated how the hyperfusogenic mutations in the F HRC domain modulate virus distribution in CNS cells. In this study, we also identified the dependence of neural cells susceptibility on both their activation state and destabilization of the virus F protein. Type I interferon (IFN-I) impaired mainly astrocytes and microglial cells permissiveness contrarily to neurons, opening a new way of consideration on the development of treatments against viral encephalitis.

## 1. Introduction

Measles virus (MeV) is a re-emerging encephalitic paramyxovirus that caused more than 200,000 deaths in 2019 [1]. Measles inclusion body encephalitis (MIBE) and subacute sclerosing panencephalitis (SSPE) are two types of rare but deadly encephalitis directly related to the presence of MeV in the brain parenchyma [2]. Part of the central nervous system invasion was recently associated with mutations in the fusion complex of MeV. This complex is composed of two surface glycoproteins: the receptor binding protein named hemagglutinin (H) and the fusion protein (F). To accomplish the entry, the H protein generally binds to high affinity receptors (CD150/SLAM and nectin-4) in order to trigger the metastable F protein, then exposing its highly hydrophobic fusion peptide that will eventually insert into the host’s cell plasma membrane [3,4,5,6]. This second conformation is unstable: F undergoes serial conformational changes, ending in the formation of a six-helix bundle composed of the heptad repeat located at its N-terminus and C-terminus (HRN and HRC) trimers that brings virus envelope and plasma membrane close enough to merge and to form the fusion pore, allowing the entry of the virus genome and replication complex [7]. Despite the absence of its known high affinity receptors, wild-type (wt) MeV seems able to enter the CNS [8]. However, most of the sequences reported from patients who died from MIBE or SSPE revealed either hypermutations of the matrix protein [9], generally abolishing its ability to interact with virus nucleocapsid and (or) surface glycoproteins or (and) mutations in the surface glycoproteins [10,11]. We have recently described that mutations often observed in the F HRC can destabilize the surface glycoprotein enough to allow the fusion in the absence of high affinity receptors [12,13,14]. Using L454W variant as a prototype of HRC-destabilized F variant, we also showed that such mutation was a mark of neuroadaptation. Hyperfusogenicity is indeed rapidly reverted in vitro, i.e., out of a CNS context, by the emergence of compensation mutations that can themselves be counter-selected when replacing the virus in the brain context [12,13].

In addition, the wt MeV differentially targets neural cells in murine organotypic cultures artificially and ubiquitously expressing CD150, thus independently of the receptor expression but depending on their activation state [14,15]. Interestingly, hamsters seem naturally more susceptible to brain infection with MeV compared with mice, which require transgenic expression of human CD150 [16].

Thus, taking the advantage of our recently developed models of hamster organotypic brain cultures [17,18], we compared, in this study, the early tropisms of wt versus L454W MeV F HRC variant and their evolution depending on the activation state. Our results show that the wt and HRC variant viruses infect all four main types of neural cells. However, the wt F virus infects few CNS cells and less differentiated oligodendrocytes, while the HRC-neuroadapted variants infect more broadly the tissue and have a tropism strongly oriented towards neurons, as observed in SSPE and MIBE patients‘ brains [19,20]. However, neuronal subtypes were not similarly susceptible to MeV infection regardless of the fusion machinery, suggesting that uncharacterized mechanisms could allow the control of viral infection.

## 2. Materials and Methods

### 2.1. Ethical Statement

All ex vivo experiments with hamsters were performed by CM and MF (accredited by the French veterinary service) according to the French National Charter on the ethics of animals. Animals used for ex vivo experiments in this study were directly euthanized by decapitation according to the AAALAC recommendations and according to French Ethical Committee (CECCAPP) regulations, accreditation # CECCAPP_ENS_2014_034.

### 2.2. Cells

293-3-46 [21,22], Vero, and Vero-SLAM/CD150 (African green monkey kidney) cells were grown in Dulbecco’s modified Eagle’s medium (DMEM; Life Technologies; Thermo Fisher Scientific, France) supplemented with 10% fetal bovine serum (FBS, Life Technologies; Thermo Fisher Scientific, France) and geneticin at 1 mg/mL (Life Technologies; Thermo Fisher Scientific, France) at 37 °C in 5% CO_2_. The 293-3-46 and Vero-SLAM/CD150 culture media were supplemented with 1 mg/mL Geneticin (Thermo Fisher Scientific, France).

### 2.3. Recombinant Virus Production and Analysis

MeV IC323-eGFP [23] is a recombinant MeV expressing the gene-encoding eGFP. All variants with the mutations T461I and L454W were generated in the MeV IC323-eGFP background using reverse genetics (using the plasmid-encoding MeV IC323-eGFP kindly provided by Yanagi, Kyushu University, Fukuoka, Japan). MeV IC323 recombinant viruses were rescued in 293-3-46 cells, as previously described [22]. Production of the viruses was performed at 32 °C. All viruses were propagated and titrated in Vero-SLAM/CD150 cells.

### 2.4. Organotypic Cerebellar Culture Preparation and Treatment Post-Infection

Organotypic cerebellar cultures (OCCs) were prepared from seven-day-old Syrian golden hamsters and maintained in culture, as detailed elsewhere [17]. Briefly, cerebella were isolated from the brains of 7-day-old hamsters and cut with a McIlwain tissue chopper (WPI-Europe) to obtain 350 µm thick progressive slices. The brain slices were then dissociated in cold Hibernate^®^-A/5 g/L D-Glucose/1x Kynurenic acid buffer and laid out on Millipore cell culture insert membranes (Millicell cell culture insert, 30 mm, hydrophilic polytetrafluoroethylene, Millipore, France). Slices were subsequently cultured in GlutaMAX minimal essential medium supplemented with 25% horse serum, 5 g/L glucose, 1% HEPES (all Thermo Fisher Scientific, France), and 0.1 mg/L human recombinant insulin (R&D Systems, France) at 37 °C in 5% CO_2_ in a humidified atmosphere. Slices from 5 hamsters were infected on the day of slicing (or 7 days later) with the indicated recombinant viruses (6 × 10^4^ plaque-forming units (PFU)/slice with wt virus and 2.5 × 10^4^ PFU/slice with L454W variant). Medium was changed at Day 1 and every 3 days.

### 2.5. Real-Time Quantitative PCR

RNA extraction, reverse transcription, and quantitative PCR were performed as previously described [24]. All results were normalized to the standard deviation (SD) for GAPDH (glyceraldehyde-3-phosphate dehydrogenase) mRNA, and the calculations were performed as previously described [24]. The primers used for MeV N quantification were MeV-N_For: 5′ GTGATCAAAGTGAGAATGAGCT 3′ and MeV-N_ReV: 5′ GCTGACCTTCGACTGTCCT 3′; and the primers used for GAPDH mRNA quantification were mGAPDH_For: 5′ GCATGGCCTTCCGTGTCC 3′ and mGAPDH_Rev: 5′ TGTCATCATACTTGGCAGGTTTCT 3′. Primers were either designed using Beacon Designer (version 8) software or chosen after validation such that their efficacy was close to 100% according to the MIQE checklist [25].

### 2.6. Immunofluorescent Staining

OCCs from seven-day-old hamsters were fixed overnight in 4% methanol-free paraformaldehyde (PFA) and washed in 1× Dulbecco’s phosphate-buffered saline (DPBS). The slices were permeabilized and blocked in 1× DPBS-3% BSA-0.3% Triton X-100 (perm and block solution) for one hour at room temperature (RT). Slices were incubated in the perm and block solution containing the primary antibodies overnight at 4 °C. After 3 washes (for 5 mins each) in 1× DPBS, slices were incubated in the perm and block solution containing the secondary antibodies (1:750) and the DAPI (1:1000) for 2h at RT. After 3 washes in 1× DPBS, slices were mounted with Fluoromount-G^®^ aqueous mounting medium (SouthernBiotech, catalog no. 0100-01). Images were taken with the LSM980 confocal microscope (Zeiss, PLATIM Lyon, France) and the CQ1 confocal microscope (Yokogawa, PLATIM Lyon, France) and analyzed using ImageJ software v1.52. Primary and secondary antibodies and dilutions used are described in Table 1 below.

### 2.7. Flow Cytometry

Twenty-four hours post-infection, neurons, astrocytes, oligodendrocytes, and microglia were labeled for flow cytometry analysis with anti-Gaba (gamma-aminobutyric acid), anti-Glast, anti-GalC (Galactocerebroside), and anti-CD68, respectively. In order to prevent damage due to the manipulation of brain cells, this antibody cocktail was specifically designed for cell-surface staining. In addition, all antibodies were already conjugated to fluorochromes, as described previously [14]. Briefly, slices were individually dissociated in 0.3 mL of 10 mg/mL of pre-activated papain (Sigma-Aldrich, France) diluted in DMEM containing 10% kynurenic acid. After 15 min of digestion at 37 °C, 0.3 mL of fetal calf serum (FCS) was added to the tube to inactivate the papain, and slices were gently dissociated by flushing. Then, cells were washed using 6 mL of washed with Neurobasal medium (Gibco; 12348017) and centrifuged at 400× *g* for 5 min. Cells were resuspended and incubated on ice for 30 min in 100 µL of antibody cocktail prepared in Neurobasal medium (Table 2). No antibody is needed to detect infected cells since they already expressed eGFP. Cells were washed as described above and fixed with 200 µL of 4% methanol-free paraformaldehyde (PFA). After 15 min incubation on ice, the cells were washed and resuspended in 100 µL of Neurobasal medium. Cells were acquired on a BD LSR Fortessa flow cytometer (BD Biosciences). Data were analyzed using FlowJo software V10.4 (TreeStar) for conventional flow cytometry. To improve the robustness of the analysis, the distribution of the infection is presented as the percentage of each stained population among eGFP-positive cells, while the contribution of each cell type to the overall infection was obtained from the percentage of infected cells in a specific subcellular population.

### 2.8. Statistical Analysis

All other statistical comparisons were performed using the Mann–Whitney U-Test. All analyses were performed in GraphPad Prism9 software.

## 3. Results

### 3.1. Invasion of Hamster Cerebellum by MeV with wt F or F HRC Variant

The susceptibility of hamster OCCs to infection by wt, F-L454W, and F-T461I (another hyperfusogenic HRC variant, [16]) viruses expressing eGFP was evaluated by daily observation. The OCCs were infected with 1000 PFU of virus at the starting day of ex vivo cultures. Viral dissemination was followed by epifluorescence microscopy (Figure 1). Four days after infection with wt virus, only a few single cells were positive for infection (detectable by eGFP fluorescence), without succeeding in infecting bystander cells (Figure 1A). On the contrary, the F-L454W- and F-T461I-bearing viruses disseminated widely in the OCCs (Figure 1B,C), and the infection was characterized by the formation of syncytia (Appendix A).

Quantification of the copies of N per μg of RNA (replication and transcription) by RT-qPCR confirmed these observations, revealing >2 log increase for the F-L454W and F-T461I viruses from 0 to 4 days post-infection (dpi) (Figure 1E,F). Thus, the mutations found in the viral sequences isolated from the brains of patients suffering from SSPE or MIBE are favorable to viral dissemination in a CNS context in the absence of a known receptor, whereas the wt virus is unable to spread in these conditions (Figure 1A,D).

### 3.2. Confocal Microscopy Analysis of MeV Early Neural Tropism

To compare the early neural tropism of hyperfusogenic variants with wt MeV, hamster OCCs were infected with 6 × 10^4^ PFU of wt or 2.5 × 10^4^ PFU of F-L454W and F-T461I viruses (Figure 2 and Appendix A). The choice of the number of PFU used for the infection was made on the basis of the lowest number of infectious particles to ensure subsequent analysis at 24 h post-infection without syncytia formation. We first qualitatively evaluated the infection of the four main CNS cell types, namely neurons, oligodendrocytes, microglial cells, and astrocytes, by confocal microscopy. We observed the infection of three cell types with the wt virus (Figure 2A–C): indeed, the Tuj-1 staining (clone of the antibody staining beta-III-Tubulin), a pan-neuronal marker, colocalized with eGFP signal, indicating neuron susceptibility to infection. Colocalization of eGFP with GFAP staining confirmed the susceptibility of astrocytes, and the few areas of eGFP/Mbp colocalization suggested the susceptibility of differentiated oligodendrocytes (but to a much lesser extent) (Figure 2B–C). On the contrary, microglial cells were not susceptible, as shown by iba-1 staining (Figure 2D). Infection with the hyperfusogenic F-L454W and F-T461I viruses led to infection of significantly more cells, but targeted cells seemed to be mainly neurons. These viruses were, nonetheless, able to also infect astrocytes, oligodendrocytes, and microglial cells (Figure 2F–H and Appendix A for hyperfusogenic F-T461I).

### 3.3. Early Neural Tropism of MeV Hyperfusogenic Variant Is Skewed toward Neurons

To quantitatively assess the distribution of infected cells with wt and hyperfusogenic viruses, a flow cytometry analysis was performed. Hamster OCCs were infected with 6 × 10^4^ pfu of MeV IC323-eGFP-F-wt or 2.5 × 10^4^ PFU of the MeV IC323-eGFP-F-L454W. At 24 h post-infection, cultures were dissociated by papain digestion, and live cells were surface-labeled with anti-GABA, anti-Glast, anti-GalC, and anti-CD68 antibodies targeting neurons, astrocytes, oligodendrocytes, and microglial cells, respectively (Figure 3A,B).

The distribution of the infected cells, i.e., the percentage of each cell type among the infected cells, for wt virus showed no significant preference for the infection of one cell type over another one, except for oligodendrocytes, which seem to be less infected (Figure 3A). Astrocytes also seemed to be slightly less infected than neurons and microglial cells. On the contrary, the F-L454W-bearing virus targeted mainly neurons, with more than 55% of infected cells being positive for GABA staining (Figure 3B). Oligodendrocytes, astrocytes, and microglial cells were represented at significantly lower levels (Figure 3B). Observations were further confirmed by counting the frequencies of each neural cell type positive to MeV infection directly on pictures taken by confocal microscopy (Figure 4, first line).

Calculations made from pictures taken of OCC from four animals first confirmed that fewer mature oligodendrocytes were infected with wt virus. Although a substantial proportion of microglial showed an eGFP signal in cytometry (Figure 3), when analyzed by confocal microscopy, none with wt virus and only 3.5% with F L454W-bearing virus had fluorescence in their entire cytoplasm, suggesting a quite high proportion of cells phagocyting infected cells at Day 7 of culture (Appendix A). Confocal microscopy also confirmed the huge increase in the proportion of infected neurons, up to 40% of stained cells and the lower representation of astrocytes with hyperfusogenic virus. Unexpectedly, a quite high proportion of cells was not stained by these four markers and showed either neuronal morphology or were positive for Olig2 staining for oligodendrocyte-engaged cells in general, suggesting a higher susceptibility of precursors for both viruses and sometimes in syncytia with F L454W-bearing virus (Appendix A). Results are summarized in Table 3. MeV-expressing hyperfusogenic F variants that have emerged in human encephalitis are endowed with enhanced propagation in the CNS cells. Because these F variants are expressed in an otherwise identical MeV genomic backbone, including an identical receptor binding protein (H), we posit that the hyperfusogenic F variants can be responsible for brain invasion in human encephalitis.

### 3.4. Astrogliosis Modifies Early Tropism of MeV

Following the slicing procedure, murine organotypic hippocampal cultures progressively express astrogliosis markers that strongly hamper the susceptibility of astrocytes and microglia to MeV infection [14]. Indeed, as previously observed in murine hippocampal slices [14], the IFN-I response in hamster cerebellum cultures increased with time, as revealed by the significant accumulation of MX1 mRNA (one of the IFN-stimulated genes) from Day 0 to Day 7 (Figure 5).

We then analyzed the potential change in the proportion of infected cell types in organotypic cultures infected with F-wt virus and F-L454W variant one week after the start of the ex vivo culture (Figure 6). Similar to that of mice organotypic cultures [14], we observed a lower susceptibility of hamster organotypic culture to infection after one week of cultures that correlated to the induction of IFN following the brain injury (Figure 6). In addition, as evaluated by confocal microscopy, most of the infected cells were neurons with both wt virus (Figure 6A) and F-L454W virus (Figure 4 middle panel and Figure 6E.) Confocal microscopy also confirmed the loss of susceptibility of microglial cells to infection with the F-L454W variant. The too strong adherence of cultures to PTFE membrane of the cell culture insert prevented their dissociation into cell suspension suitable for cytometry. However, we also observed numerous microglial cells which were harboring only dotted eGFP signal in both conditions, suggesting that the positivity is due more to phagocytosis than infection with active replication, as observed for infection made at Day 0 (Appendix A). We also noticed fewer unstained infected cells, suggesting that precursors had differentiated during the week of culture.

The overall loss of susceptibility of the OCCs infected after 7 days post-slicing suggests that even an MeV-bearing unstable fusion machinery cannot fully overcome an antiviral response developed in this brain structure.

### 3.5. IFN-I Pretreatment Alters the Permissiveness of Neural Cells to MeV Infection

In mice organotypic brain cultures, the astrogliosis and its consequences on astrocytes and microglia susceptibility are directly related to IFN-I signaling pathway [14]. We notably observed in hippocampal slices that permissiveness of these two cell types decreases over time concomitantly to the overexpression of IFN-I and the development of astrogliosis. Thus, we next assessed whether astrogliosis and the associated changes in susceptibility could be mimicked by an IFN-I pretreatment (Figure 7). As expected, IFN-I reduced the global susceptibility of the OCC to both wt and L454W variant virus (Figure 7A), from 562 to 141 and from 1016 to 758 positive cells of eGFP per OCC, respectively. Moreover, as observed when the OCCs were infected 7 days after culture preparation, cytometry analysis showed that for F-L454W, and even more drastically for wt virus, the proportion of infected astrocytes and microglial cells among all eGFP positive cells was significantly reduced compared with neurons that increased (also significantly) and oligodendrocytes that remained stable (Figure 7B). The increased proportion of neurons compared with other cell types was also confirmed by calculations made from confocal microscopy pictures (Figure 4 lower panels). Results further indicate that neurons (at least for one part) do not seem able to enter an efficient antiviral state upon IFN-I response and remain permissive.

### 3.6. MeV Does Not Infect All Neurons

As shown above, neurons were among the most infected cells in all tested conditions. However, the absence of colocalization of eGFP with the CB28K marker of Purkinje and Golgi neurons indicates that these two neuronal subtypes are very poorly permissive to MeV infection. The hyperfusogenicity did not allow overcoming the resistance of these cells. This suggests that the entry step may not be the limiting factor, and virus replication cycle may not be initiated in these particular neurons (Figure 8).

## 4. Discussion

Mutations in the HRC region of the MeV F protein that confer hyperfusogenicity have been associated with SSPE and MIBE [2,6,11,12,27]. We have highlighted that hyperfusogenic F variants, such as F-L454W and F-T461I, lead to intense spread and significant increase in viral N RNA copies in hamster organotypic brain cultures, while the wt virus can infect only a few cells with very little to no viral dissemination.

To go further, we comparatively evaluated the permissiveness of the CNS cells to both wt and hyperfusogenic virus, and we thus assessed whether F instability could also affect the virus tropism. Here, as formerly suggested, MeV can be found in all cell types from the brain [14,15]. However, differentiating oligodendrocytes seemed less susceptible to the infection, and positivity of microglia more likely reflects phagocytosis of nearby infected dying cells. Hyperfusogenic MeV infected all four cell types with a significant dominance for neurons and oligodendrocytes precursors. Such results suggest that tropism is poorly affected since both viruses can infect all cell types except a few microglial cells. However, hyperfusogenicity increases the inclination to spread faster into the cells, producing less (and eventually responding less to) IFN-I.

Moreover, as commonly observed in MeV encephalitis, the slicing procedure also causes an astrogliosis fully efficient thanks to the IFN-I pathway [14]. Thus, we artificially induced an IFN-I response by culturing the OCCs for seven days or by stimulating the OCCs with a low dose of IFN-I known to be activating in vitro. The astrogliosis has been previously shown to reduce the susceptibility of astrocytes and microglia to wt MeV in OCCs from mice artificially expressing SLAM F1. wt MeV itself also induces an IFN response in mice brain cultures [14]. Here we confirm the post-slicing IFN induction in the hamster model. We additionally highlighted that, in the presence of IFN-I response/stimulation, the infection of the astrocytes and microglia were strongly reduced, with little to no impact on the neurons and oligodendrocytes infection. Trying to mimic the IFN-I effect on astrogliosis development following injury, we also observed that the neural tropism proportion was altered and became oriented mainly to neurons, suggesting the lower ability of these cells to enter an antiviral state upon IFN-I exposure. Astrocytes (and microglia for the F-L454W variant) permissiveness was significantly lower after astrogliosis phenomenon or IFN-I stimulation. How such an observation could explain MeV evolution in the brain remains to be further explored. To what extent wt MeV could evolve into a hyperfusogenic phenotype, such as F L454W under IFN-I pressure, will also need further investigation. We demonstrated that both F instability and IFN-I could be at the origin of the higher propensity to spread in neurons (and potentially neuronal and oligodendrocyte precursors) as always observed in patients compared with differentiated oligodendrocytes, astrocytes, and microglia that are less commonly observed.

Strikingly, for MeV wt F, the IFN-I treatment recapitulated the phenotype observed with the MeV F-L454W variant in terms of neuronal tropism: wt MeV infects mainly neurons (and oligodendrocytes) in stimulated slices. This observation suggests a link among F hyperfusogenicity, IFN-I induction, and cell permissiveness. Mechanistically, we propose that the instability of the F and, thus, its high propensity to be triggered and mediate virus-cell membrane fusion are perceived by the cell as a strong danger signal, which induces the IFN pathway, as posited for the herpes simplex virus, for example [28]. This notion is further supported by the fact that syncytia formation (cell–cell fusion) induced by MeV is a potent inducer of IFN-I response [29]. The IFN induction resulting from rapid and strong fusion ability of hyperfusogenic MeV puts the responder cells, such as astrocytes and microglia, in an antiviral state that renders them less permissive. On the other hand, non-responder cells, such as neurons and oligodendrocytes, would remain permissive. Interestingly, regardless the IFN-I, no infection has been found in neurons expressing CB28K, neither for wt F nor for F-L454W MeV.

In humans, in post-mortem analyses from MIBE or SSPE patients’ brains, viral antigens are generally found in neurons, and more occasionally in astrocytes, oligodendrocytes, and microglia [2,14]. Whether the virus enters the CNS using immune cells as a Trojan horse or peripheral nerves, endothelial cells or simply neurons from the olfactory bulb cannot be addressed using these cultures. However, they allow proposing that both wt or hyperfusogenic F-bearing viruses can trigger an IFN-I response and an astrogliosis unbalancing the cell susceptibility to privilege infection of neurons (and oligodendrocytes at least precursors). In the same way, hyperfusogenicity strongly increases the proportion of virus-infecting neurons. In humans, infected oligodendrocytes are often localized close to infected neurons [19,20,30], suggesting a secondary infection of the oligodendrocytes via the axons that could be investigated by electrophysiology and spinning confocal microscopy. The proportion of active infection of microglial cells versus positives to eGFP signal-reflecting phagocytosis will need to be further investigated as well. Hyperfusogenicity has not allowed the virus to infect Purkinje cells. In humans, their infection was not reported, further confirming that permissiveness to MeV does not only rely on the entry step, as we previously described in murine SLAM transgenic organotypic cultures [14]. Further investigation regarding resistance of Purkinje neurons may help in highlighting key factors of the infection.

It remains unclear to what extend the infection of the CNS is common or not. However, the success of the neuroinvasion may rely on the combination of the virus’ ability to enter enough in the most susceptible cells (i.e., neurons and oligodendrocytes precursors) either thanks to F-destabilizing mutations combined with IFN-I response or because of the lack of appropriate antiviral response, as observed in numerous MIBE cases.

Future directions could be to evaluate whether neurons (and oligodendrocytes) could constitute a somehow more mutagenic environment, potentially at the origin of the virus evolution to hyperfusogenicity under selective pressure of the currently used treatments of viral infection, such as IFN-I, ribavirin, or Remdesivir (GS-5734) [31]. In this hypothesis, antiviral drugs may strongly decrease the permissiveness of microglia and astrocytes—at the same time and as a side effect favoring infection of neurons—since they would remain the only permissive cells. This would be equivalent to reducing the “dilution” of the virus in the brain and would eventually be detrimental because the virus would be in a mutation-prone context. In rare cases, long term IFN-α treatment can stabilize the clinical symptom of SSPE for years and is often associated with ribavirin [30,32,33,34]. However, the emergence of additional mutations in response to these treatments has not been investigated in the literature. Nevertheless, such notions will become essential for future development of appropriate treatments of patients hospitalized with viral encephalitis.

## Figures and Tables

**Figure 1 viruses-15-00229-f001:**
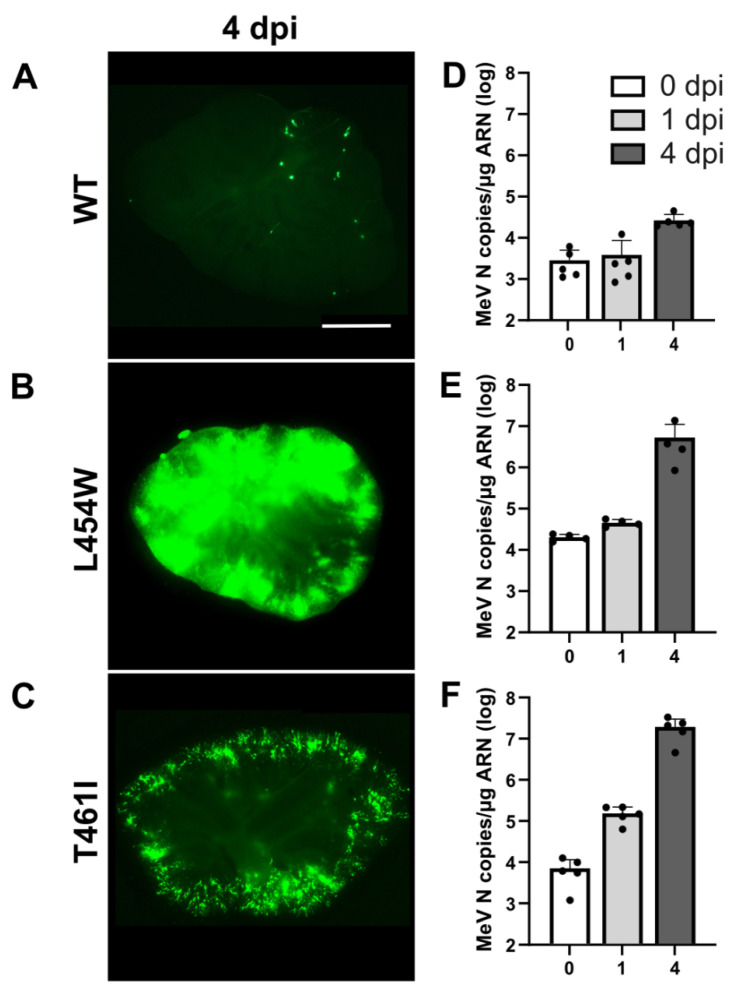
Viral dissemination of MeV-bearing F variant proteins in hamster organotypic cerebellar cultures (OCCs). OCCs were infected on the day of preparation with 1000 PFU of either wild-type F bearing MeV or those showing variants in F. (**A**–**C**) Photos of the OCCs 4 days post-infection. Cells expressing eGFP are evidence of viral replication. Photos were taken by the Nikon Eclipse Ts2R microscope at 200 milliseconds (ms) exposure for A and at 50 ms exposure for B and C and reconstructed using the Stitching plug-in with the ImageJ Software [26]. (**D**–**F**) Quantification of the copies of N per μg of RNA for each virus at 0, 1, and 4 days post-infection (n = 5). Scale bar = 1 mm.

**Figure 2 viruses-15-00229-f002:**
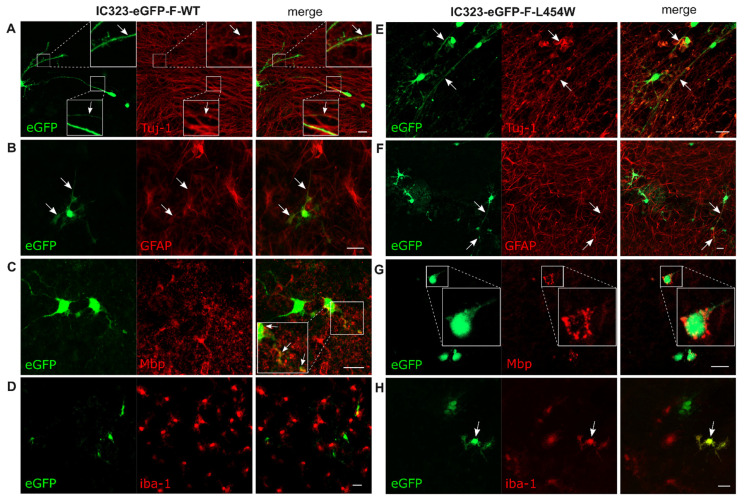
Early tropism of wild-type measles virus and MeV IC323-eGFP-F-L454W infections in hamster cerebellar organotypic cultures (OCCs). (**A**–**D**) OCCs were infected with 6 × 10^4^ PFU of the MeV IC323-eGFP-F-wt on the day of preparation. OCCs were fixed in 4% formaldehyde at 48 h post-infection. (**A**,**E**) Immunofluorescent (IF) staining of tubulin present in neurons (Beta-III-Tubulin, clone Tuj-1). (**B**,**F**) IF staining of astrocytes (GFAP) and granular neurons. (**C**,**G**) IF staining of microglial cells (iba-1). (**D**,**H**) IF staining of oligodendrocytes (Mbp). **(E**–**H)** In IF, green fluorescence corresponds to infected cells where the virus expresses eGFP. The images were acquired by confocal microscopy and analyzed with the ImageJ software. Arrows point to areas of colocalization between infected cells and the staining of interest. Scale bar = 20 µm.

**Figure 3 viruses-15-00229-f003:**
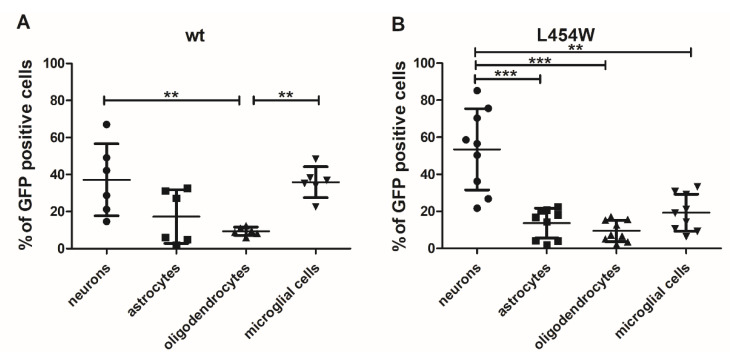
Evaluation of early tropism of MeV-IC323-eGFP-F-wt and MeV-IC323-eGFP-F-L454W infection in hamster cerebellar organotypic cultures (OCCs) by flow cytometry. OCCs were infected (**A**) 6 × 10^4^ pfu of MeV IC323-eGFP-F-wt expressing eGFP virus to get enough signal for flow cytometry analysis or (**B**) with 2.5 × 10^4^ PFU of the MeV IC323-eGFP-F-L454W. At 24 h after infection, neurons, astrocytes, oligodendrocytes, and microglial cells were labeled by extracellular staining for analysis by flow cytometry using antibodies conjugated to fluorochromes: anti-Gaba, anti-Glast, anti-GalC, and anti-CD68, respectively. Data represent the distribution of infection, i.e., the percentage of each cell type among infected cells. * *p* < 0.05; ** *p* < 0.01; *** *p*< 0.001.

**Figure 4 viruses-15-00229-f004:**
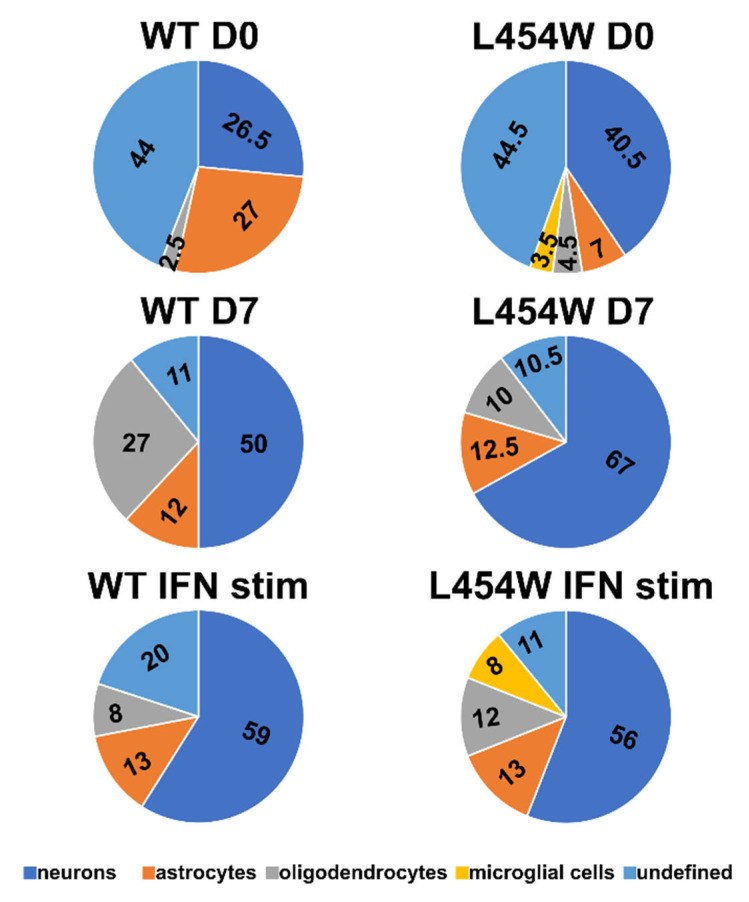
Evaluation of early tropism of MeV-IC323-eGFP-F-wt and MeV-IC323-eGFP-F-L454W infection in hamster cerebellar organotypic cultures by confocal spinning disk microscopy (pictures obtained from at least 4 animals). Upper panel shows the distribution of the eGFP when infection is performed at Day 0. Middle panel shows the distribution when the infection is performed at Day 7 of culture. Lower panel shows the distribution when the infection is performed at Day 0, 4 h after stimulation with type 1 IFN (1000 units/OCC). Undefined means diffused eGFP in the cytoplasm without cell marker staining. Neurons, astrocytes, oligodendrocytes, and microglial cells were stained with anti-Tuj1, anti-GFAP, anti-Mbp, and anti-iba-1, respectively.

**Figure 5 viruses-15-00229-f005:**
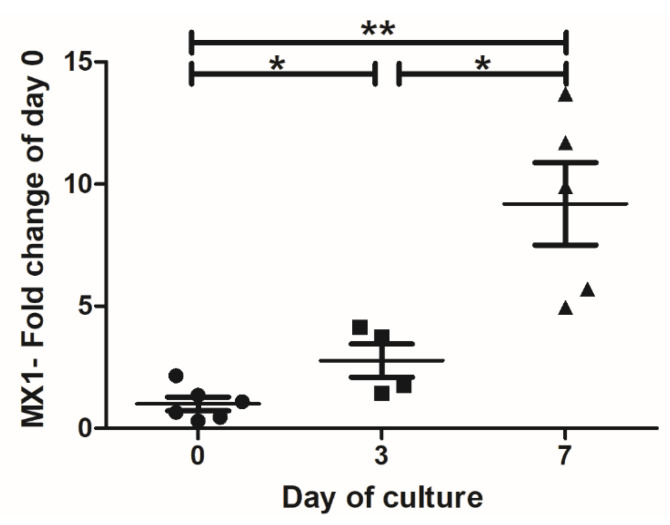
Impact of cell activation on tropism of MeV variants infection after 7 days of culture. mRNA expression level of MX1 (Myxovirus Resistance 1) over time after the slicing procedure of the hamster cerebellum organotypic cultures. mRNA copies per µg of total RNA were quantified by RT-qPCR and normalized to the variation of the amounts of GAPDH mRNA. Fold changes are relative to the number of copies of mRNA compared with Day 0. Error bars represent SD. * *p* < 0.05; ** *p* < 0.01.

**Figure 6 viruses-15-00229-f006:**
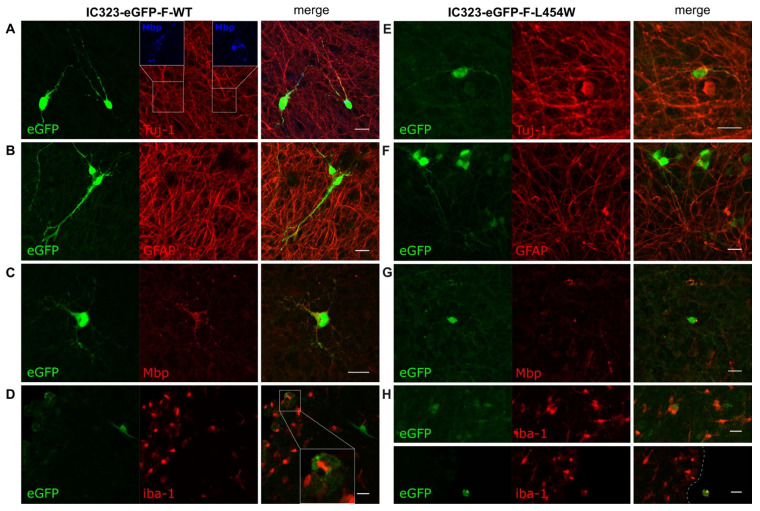
Impact of cell activation on tropism of MeV wt and MeV F L454W tropism after 7 days of culture. **(A**–**D)** The OCCs were infected with 6 × 10^4^ PFU of the MeV IC323-eGFP-F-wt or (**E–H**) 2.5 × 10^4^ PFU of the MeV IC323-eGFP-F-L454W virus after 7 days of culture and fixed in 4% formaldehyde 24 h post-infection. (**A**,**E**) Immunofluorescent staining of tubulin present in neurons (Beta-III-Tubulin, clone Tuj-1), (**B**,**F**) of astrocytes (GFAP), (**C**,**G**) of oligodendrocytes (Mbp), and (**D**,**H**) of microglial cells (iba-1). Line shows the edge of the OCC. Green fluorescence corresponds to infected cells with virus expressing eGFP. The images were acquired by confocal microscopy and analyzed with the ImageJ software. Scale bar = 20 µm.

**Figure 7 viruses-15-00229-f007:**
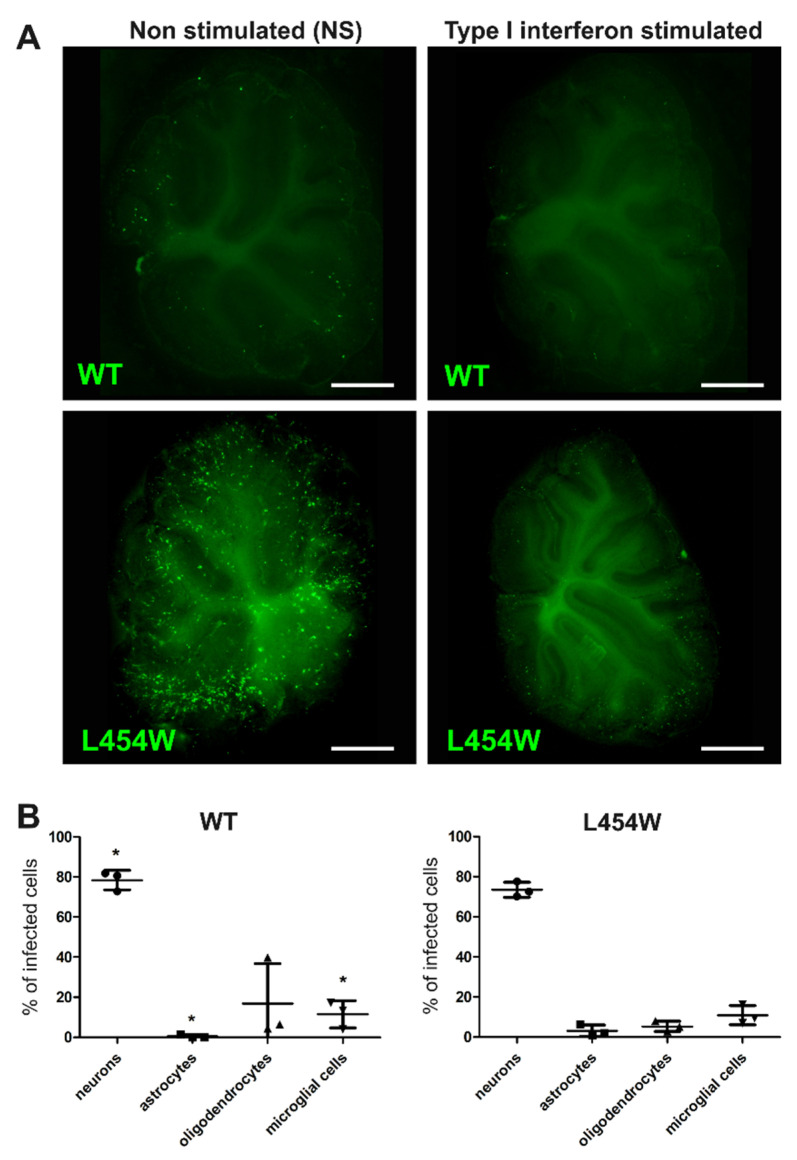
Type I interferon pre-stimulation impact MeV tropism in the brain. (**A**) The hamster organotypic cerebellar cultures (OCCs) were treated (or not) for 4h with 1000 units of IFN-I. Then, OCCs were infected with either 6 × 10^4^ PFU of the MeV IC323-eGFP-F-wt or 2.5 × 10^4^ PFU of the MeV IC323-eGFP-F-L454W on the day of the section. Photos were taken by the Nikon Eclipse Ts2R microscope 24 h after infection. (**B**) At 24 h after infection, treated infected OCCs were dissociated, and neurons, astrocytes, oligodendrocytes, and microglial cells were labeled by extracellular staining for analysis by flow cytometry using antibodies conjugated to fluorochromes: anti-Gaba, anti-Glast, anti-GalC, and anti-CD68, respectively. Data represent the distribution of infection, i.e., the percentage of each cell type among infected cells was quantified. * *p* < 0.05. Scale bar = 1 mm.

**Figure 8 viruses-15-00229-f008:**
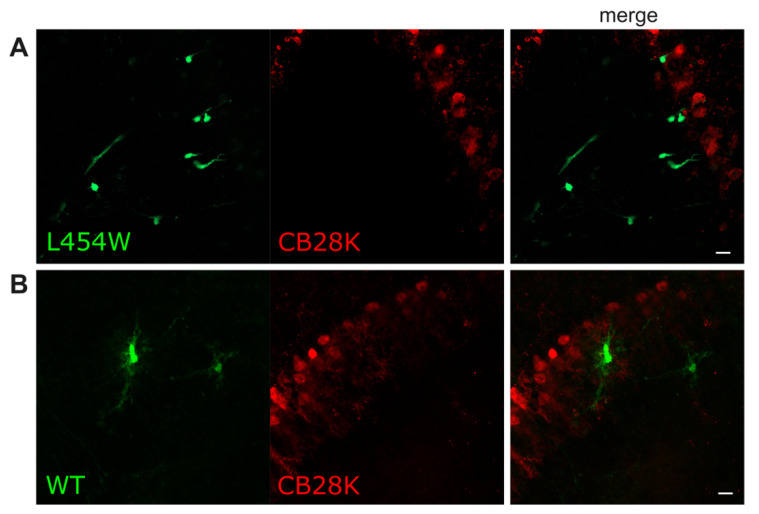
Purkinje and Golgi neurons resist MeV infection in the brain. The hamster organotypic cerebellar cultures were infected with 10^4^ PFU of (**A**) the MeV IC323-eGFP-F-L454W or (**B**) the MeV IC323-eGFP-F-wt on the day of the section and fixed in 4% formaldehyde 24 h post-infection. Immunofluorescent (IF) staining of Purkinje and Golgi neurons (CB28K) are shown in red. Green fluorescence corresponds to infected cells where virus expresses eGFP. The images were analyzed with the ImageJ software. Scale bar = 20 µm.

**Table 1 viruses-15-00229-t001:** Antibodies used for immunofluorescent staining.

Antibody	Dilution Used	Reference
Mouse monoclonal anti-Myelin Basic Protein (clone MBP101); Abcam, France	1/200	Cat# ab62631
Rabbit polyclonal anti-Glial Fibrillary Acidic Protein (GFAP); Agilent Technologies, Dako, France	1/500	Cat# Z0334
Rabbit anti-beta III Tubulin (Tuj-1); Abcam, France	1/500	Cat# ab18207
Goat polyclonal anti-Olig2; R&D system, France	1/200	Cat# AF2418
Rabbit polyclonal anti-iba-1; Wako, USA	1/200	Cat# 016-20001
Goat polyclonal anti-calbindin D-28K (CB28K); Invitrogen, France	1/500	Cat# PA5-46936
Alexa FluorTM 555 Donkey anti-mouse IgG (H+L); ThermoFisher Scientific, France	1/500	Cat# A31570
Alexa FluorTM 555 Donkey anti-rabbit IgG (H+L); ThermoFisher Scientific, France	1/500	Cat# A31572
Alexa FluorTM 647 Donkey anti-mouse IgG (H+L); ThermoFisher Scientific, France	1/500	Cat# A31571
Alexa FluorTM 647 Donkey anti-goat IgG (H+L); ThermoFisher Scientific, France	1/500	Cat# A21447

**Table 2 viruses-15-00229-t002:** Antibodies used for extracellular staining for flow cytometry.

Cell Population	Antibody	Fluorochrome	Reference	Dilution
Neurons	Monoclonal Anti-GABA A Receptor	PE/Atto 594	Merk/sigma; SAB5202234-100UG	1/100
Astrocytes	Anti GLAST (ACSA-1), anti-human/mouse/rat	PE	Miltenyi; 130-118-344	1/10
Oligodendrocytes	Clone mGalC Anti-Galactocerebroside	Alexa fluor 647	Merk/sigma; MAB342-AF647	1/100
Microglial cells	CD68 Rat-Anti-Mouse	BV421	BD; 566388	1/50

**Table 3 viruses-15-00229-t003:** Infection of different CNS cell types with measles virus variants early after infection.

	IC323-eGFP-F-wt	IC323-eGFP-F-L454W
Tubulin-positive neurons (Tuj-1)	++	++++
Astrocytes (GFAP)	++	+
Oligodendrocytes (Mbp)	+	+
Microglial cells (iba-1)	−	+
Oligodendrocytes-engaged cells (Olig2)	++	+++
CB28K-positive neurons de (Purkinje and Golgi cells)	−	−

(+) Observation of colocalization between infected cells and cells of interest; (−) No colocalization observed.

## Data Availability

All data are available on request from the authors.

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
