# Peer review of "Early Permissiveness of Central Nervous System Cells to Measles Virus Infection Is Determined by Hyperfusogenicity and Interferon Pressure"

_viruses, 2023, doi:10.3390/v15010229_

Round 1
Reviewer 1 Report
General comments.
This manuscript presents data on MeV infection in hamster organotypic brain cultures. The authors reported that the MeV infected all cell types of neurons, oligodendrocytes, microglial cells, and astrocytes, but cell type-preferences of MeV infection differed depending on the stability of MeV F protein and the IFN-I response.
This reviewer appreciates the effort of the work presented and the significance of ex vivo analysis of MeV neurotropism but has serious concerns over the experimental method used in this study. Thus, in the present state, this manuscript is inappropriate for publication.
Major
1. This paper shows colocalization of MeV infected cells and cell-type specific markers in several figures. However, this reviewer disagrees that white arrows in Figs. 2D and 3C indicate colocalization of eGFP and iba-1, and eGFP and MBP, respectively, because the shapes of stained regions are totally different. They might indicate overlapping cells of different types.
2. The sums of percentages of Fig. 4 and Fig. 8B are obviously greater than 100%. In Fig. 8B they reach even 160%. This reviewer does not feel it ‘slightly’ greater. Thus, the reliability of the data obtained from this method are not significant enough to discuss the distribution of infected cells.
3. In Fig. 4, the numbers of samples are different between WT (A) and L454W (B) MeVs. They must affect the result of statistical analysis due to different sensitivities for significance. The sample numbers should be the same if the authors statistically compare cell-type preference of WT MeV with that of L545W MeV.
Minor
1. Page 5, Line 178
This reviewer does not agree that from the images in Fig. 1B and C the authors could distinguish syncytia from crowding infected cells.
2. Page 7, lines 200-207
Please correct references for figures.
e.g. line 204 Figure 2C, D to Figure 2B
3. As the oligodendrocyte markers, Olig-2 was used in Fig. 2 (WT), but MBP was used in Fig. 3 (L454W). If there is no specific reason, please use the same marker for both experiments.
Author Response
Reviewer 1
This manuscript presents data on MeV infection in hamster organotypic brain cultures. The authors reported that the MeV infected all cell types of neurons, oligodendrocytes, microglial cells, and astrocytes, but cell type-preferences of MeV infection differed depending on the stability of MeV F protein and the IFN-I response.
This reviewer appreciates the effort of the work presented and the significance of ex vivo analysis of MeV neurotropism but has serious concerns over the experimental method used in this study. Thus, in the present state, this manuscript is inappropriate for publication.
Major
- This paper shows colocalization of MeV infected cells and cell-type specific markers in several figures. However, this reviewer disagrees that white arrows in Figs. 2D and 3C indicate colocalization of eGFP and iba-1, and eGFP and MBP, respectively, because the shapes of stained regions are totally different. They might indicate overlapping cells of different types.
=>We agree with the reviewer that the shape of eGFP does not always fill the whole microglial cell cytoplasm, that is why we mentioned in the text and in supplementary figure that part of the positive cells might be in fact result from phagocytosis of part of infected cells and not active replication. We have more insisted on that point and provided new pictures of both microglial cells and oligodendrocytes showing susceptibility of precursors and low but existing susceptibility of oligodendrocytes in differentiation (Spotted MBP staining fitting with cell shape, in cytoplasm but not in the nucleus).
- The sums of percentages of Fig. 4 and Fig. 8B are obviously greater than 100%. In Fig. 8B they reach even 160%. This reviewer does not feel it ‘slightly’ greater. Thus, the reliability of the data obtained from this method are not significant enough to discuss the distribution of infected cells.
=>In the initial submission we have tried to get the maximum of cells with low intensity to make sure we were not excluding cell poorly replicating the virus since it is not the question addressed in this study. This can lead to inclusion of highly autofluorescent cells and cells such as myelinated neurons still catching markers from both neurons (GABA) and oligodendrocytes (GalC). We have reanalyzed the data to remove both element taking the risk of loosing poorly eGFP positive cells and came thus down to close to 100%. We have also added the contribution of each cell type to the initial infection as formerly described (in Welsch et al J Virol 2019) to the analysis in order to get a clearer view.
We also provided a second method of calculation based on confocal microscopy to confirm observation made in Flow cytometry and deepen the analysis.
- In Fig. 4, the numbers of samples are different between WT (A) and L454W (B) MeVs. They must affect the result of statistical analysis due to different sensitivities for significance. The sample numbers should be the same if the authors statistically compare cell-type preference of WT MeV with that of L545W MeV.
=>We agree and that is why we did not directly compare WT and L454W in statical analysis; Analysis was done within each virus condition ensuring the exclusion of such a bias.
Minor
- Page 5, Line 178
This reviewer does not agree that from the images in Fig. 1B and C the authors could distinguish syncytia from crowding infected cells.
=>We agree pictures quality could be too low, thus we have provided a better and clearer picture in supplementary S2.
- Page 7, lines 200-207
Please correct references for figures.
e.g. line 204 Figure 2C, D to Figure 2B
=>Done, we have a new figure 2.
- As the oligodendrocyte markers, Olig-2 was used in Fig. 2 (WT), but MBP was used in Fig. 3 (L454W). If there is no specific reason, please use the same marker for both experiments.
=>no specific reason at that time, MBP stains more differentiated (or in differentiation) oligodendrocyte, and picture was just better and allows eventually visualizing myelin gain around neurons; We have provided equivalent staining in both wt and L454W.
Reviewer 2:
The manuscript "Early permissiveness of central nervous system cells to measles virus infection is determined by hyperfusogenicity and interferon pressure" highlights an interesting but relatively understudied aspect of the permissiveness of central nervous system cells to measles virus.
The authors show that hyperfusogenic mutations in the F HRC domain modulate the distribution of the virus in CNS cells and highlight a differential impact of type 1 interferon, opening the way to the identification of new therapeutic alternatives. All the work was carried out using an original model of organo-typic hamster brain cultures (OCC). The topic is timely and interesting, the data is promising, and most conclusions are supported by the data.
=>We thank the reviewer for the very encouraging comments and constructive suggestions.
Major points
For the early analysis of MeV tropism, OCCs were infected with 104 PFUs of wt, F-L454W and F-T461I and shown in Figure 2, Figure 3 and Figure S1, the independent separation of results into 3 figures makes comparison difficult, figures 2 and 3 could be merged to facilitate analysis.
For cell subpopulation analysis, Olig-2 was used for experiments with the wt strain and F-T461I, and MDP for F-L454W. What is the reason for the analysis change?
=>We thank the reviewer for the observation, we have added the equivalent staining and a new picture in supplemental material to differentiate precursors from differentiated cells.
Results are expressed qualitative analysis, a quantification of the frequency of differences in MeV-infected subpopulations would be important to support the results.
=>We have repeated multiple times the experiments to get better overview of the frequencies which are now in the manuscript as a new figure.
The authors then carried out the quantitative analysis by flow cytometry of the infection of the different cell subpopulations. For wt 105 PFU were used while for L454W 104. Can the analysis be performed with the same PFU for both strains? Could this difference have an impact on cellular tropism?
=>We apology. We in fact used 2.5x104 PFU of L454W virus and 6x104 PFU of wt virus. We have corrected and homogenized in the manuscript. Unfortunately, using 2.5x 104 PFU of wt does not allow getting enough infection to visualize positive cells by flow cytometry (or too few to avoid bias) alternatively putting more than 2.5x 104 PFU of L454W variant rapidly leads to syncytia formation completely biasing the analysis. We have added a sentence in the text to justify this choice in paragraph 3.2.
For Figures 6 and 7, as for Figures 2 and 3, it would be preferable to combine them in order to facilitate the analysis. In addition, the frequencies of infected cells have been indicated in Figure 6. What is the methodology used for this quantification? How many times has this experience been carried out.
=> Calculation were made by scanning the whole slices to find both positive staining out of n≥4 animals. We obtain the percentage of stained cells among total EGFP positive cells. The main risk is that sometimes positive cells in the most inner part of the slice may be less stained.
For Figure 7, frequencies are missing.
Figure 6 and 7 have been pooled and a new figure with frequencies has been added (figure 4).
The analysis of the impact of treatment with type I interferons on the infection of the different subpopulations carried out in Fig. 8 requires further investigation to substantiate the presented results.
In fact, the analysis by flow cytometry seems to be carried out only after treatment with type I interferon. However, the results presented in figure 4 already show a "low" level of infection for the astrocytes and the microglial cells. The comparison with and without treatment with type I interferon is therefore important to show the decrease in the infection of these two cellular subpopulations and to confirm their antiviral status upon IFN-1 response.
=>We thank the reviewer for the suggestion. We have completed the analyses with confocal microscopy and added frequencies in Figure 4.
Moreover, for Figure 8B, the total percentage of cells is close to 170/180% illustrating the problem of double/triple positivity of cells which could impact even slightly the proportions of the different cell subpopulations.
=>We agree with reviewer’s observation. As mentioned to reviewer 1, we tried to get a maximum of information by integrating low positive cells. The side effect of that is that we both have double positive cells (mainly Neurons/oligodendrocytes) but also some of the cells with more autofluorescence. We have thus increased the stringency of our threshold to come back closer to 100%, taking this time the risk of loosing knowing that microscopy analysis would complete the observation.
Minor points
Fig 2, 3 and S1, it would be preferable to use identical magnifications as well as the same order of representation
=>We apology. We tried to play with magnification to show better colocalization or not. Frequencies are now represented as a separated figure to complete observations.
Legend for figure 4: inversion between A and B
=>We have modified in the new figure 3 legend.
Fig 6 and 7 Tuj is shown instead of tuj-1
=>we have corrected the figures.
Fig 8A, global quantification of infected cells would provide quantitative support for the results
+We have added the values in the paragraph 3.5.

Reviewer 2 Report
The manuscript "Early permissiveness of central nervous system cells to measles virus infection is determined by hyperfusogenicity and interferon pressure" highlights an interesting but relatively understudied aspect of the permissiveness of central nervous system cells to measles virus.
The authors show that hyperfusogenic mutations in the F HRC domain modulate the distribution of the virus in CNS cells and highlight a differential impact of type 1 interferon, opening the way to the identification of new therapeutic alternatives. All the work was carried out using an original model of organo-typic hamster brain cultures (OCC). The topic is timely and interesting, the data is promising, and most conclusions are supported by the data.
Major points
For the early analysis of MeV tropism, OCCs were infected with 104 PFUs of wt, F-L454W and F-T461I and shown in Figure 2, Figure 3 and Figure S1, the independent separation of results into 3 figures makes comparison difficult, figures 2 and 3 could be merged to facilitate analysis.
For cell subpopulation analysis, Olig-2 was used for experiments with the wt strain and F-T461I, and MDP for F-L454W. What is the reason for the analysis change?
Results are expressed qualitative analysis, a quantification of the frequency of differences in MeV-infected subpopulations would be important to support the results.
The authors then carried out the quantitative analysis by flow cytometry of the infection of the different cell subpopulations. For wt 105 PFU were used while for L454W 104. Can the analysis be performed with the same PFU for both strains? Could this difference have an impact on cellular tropism?
For Figures 6 and 7, as for Figures 2 and 3, it would be preferable to combine them in order to facilitate the analysis. In addition, the frequencies of infected cells have been indicated in Figure 6. What is the methodology used for this quantification? How many times has this experience been carried out.
For Figure 7, frequencies are missing.
The analysis of the impact of treatment with type I interferons on the infection of the different subpopulations carried out in Fig. 8 requires further investigation to substantiate the presented results.
In fact, the analysis by flow cytometry seems to be carried out only after treatment with type I interferon. However, the results presented in figure 4 already show a "low" level of infection for the astrocytes and the microglial cells. The comparison with and without treatment with type I interferon is therefore important to show the decrease in the infection of these two cellular subpopulations and to confirm their antiviral status upon IFN-1 response.
Moreover, for Figure 8B, the total percentage of cells is close to 170/180% illustrating the problem of double/triple positivity of cells which could impact even slightly the proportions of the different cell subpopulations.
Minor points
Fig 2, 3 and S1, it would be preferable to use identical magnifications as well as the same order of representation
Legend for figure 4: inversion between A and B
Fig 6 and 7 Tuj is shown instead of tuj-1
Fig 8A, global quantification of infected cells would provide quantitative support for the results
Author Response
Reviewer 1
This manuscript presents data on MeV infection in hamster organotypic brain cultures. The authors reported that the MeV infected all cell types of neurons, oligodendrocytes, microglial cells, and astrocytes, but cell type-preferences of MeV infection differed depending on the stability of MeV F protein and the IFN-I response.
This reviewer appreciates the effort of the work presented and the significance of ex vivo analysis of MeV neurotropism but has serious concerns over the experimental method used in this study. Thus, in the present state, this manuscript is inappropriate for publication.
Major
- This paper shows colocalization of MeV infected cells and cell-type specific markers in several figures. However, this reviewer disagrees that white arrows in Figs. 2D and 3C indicate colocalization of eGFP and iba-1, and eGFP and MBP, respectively, because the shapes of stained regions are totally different. They might indicate overlapping cells of different types.
=>We agree with the reviewer that the shape of eGFP does not always fill the whole microglial cell cytoplasm, that is why we mentioned in the text and in supplementary figure that part of the positive cells might be in fact result from phagocytosis of part of infected cells and not active replication. We have more insisted on that point and provided new pictures of both microglial cells and oligodendrocytes showing susceptibility of precursors and low but existing susceptibility of oligodendrocytes in differentiation (Spotted MBP staining fitting with cell shape, in cytoplasm but not in the nucleus).
- The sums of percentages of Fig. 4 and Fig. 8B are obviously greater than 100%. In Fig. 8B they reach even 160%. This reviewer does not feel it ‘slightly’ greater. Thus, the reliability of the data obtained from this method are not significant enough to discuss the distribution of infected cells.
=>In the initial submission we have tried to get the maximum of cells with low intensity to make sure we were not excluding cell poorly replicating the virus since it is not the question addressed in this study. This can lead to inclusion of highly autofluorescent cells and cells such as myelinated neurons still catching markers from both neurons (GABA) and oligodendrocytes (GalC). We have reanalyzed the data to remove both element taking the risk of loosing poorly eGFP positive cells and came thus down to close to 100%. We have also added the contribution of each cell type to the initial infection as formerly described (in Welsch et al J Virol 2019) to the analysis in order to get a clearer view.
We also provided a second method of calculation based on confocal microscopy to confirm observation made in Flow cytometry and deepen the analysis.
- In Fig. 4, the numbers of samples are different between WT (A) and L454W (B) MeVs. They must affect the result of statistical analysis due to different sensitivities for significance. The sample numbers should be the same if the authors statistically compare cell-type preference of WT MeV with that of L545W MeV.
=>We agree and that is why we did not directly compare WT and L454W in statical analysis; Analysis was done within each virus condition ensuring the exclusion of such a bias.
Minor
- Page 5, Line 178
This reviewer does not agree that from the images in Fig. 1B and C the authors could distinguish syncytia from crowding infected cells.
=>We agree pictures quality could be too low, thus we have provided a better and clearer picture in supplementary S2.
- Page 7, lines 200-207
Please correct references for figures.
e.g. line 204 Figure 2C, D to Figure 2B
=>Done, we have a new figure 2.
- As the oligodendrocyte markers, Olig-2 was used in Fig. 2 (WT), but MBP was used in Fig. 3 (L454W). If there is no specific reason, please use the same marker for both experiments.
=>no specific reason at that time, MBP stains more differentiated (or in differentiation) oligodendrocyte, and picture was just better and allows eventually visualizing myelin gain around neurons; We have provided equivalent staining in both wt and L454W.
Reviewer 2:
The manuscript "Early permissiveness of central nervous system cells to measles virus infection is determined by hyperfusogenicity and interferon pressure" highlights an interesting but relatively understudied aspect of the permissiveness of central nervous system cells to measles virus.
The authors show that hyperfusogenic mutations in the F HRC domain modulate the distribution of the virus in CNS cells and highlight a differential impact of type 1 interferon, opening the way to the identification of new therapeutic alternatives. All the work was carried out using an original model of organo-typic hamster brain cultures (OCC). The topic is timely and interesting, the data is promising, and most conclusions are supported by the data.
=>We thank the reviewer for the very encouraging comments and constructive suggestions.
Major points
For the early analysis of MeV tropism, OCCs were infected with 104 PFUs of wt, F-L454W and F-T461I and shown in Figure 2, Figure 3 and Figure S1, the independent separation of results into 3 figures makes comparison difficult, figures 2 and 3 could be merged to facilitate analysis.
For cell subpopulation analysis, Olig-2 was used for experiments with the wt strain and F-T461I, and MDP for F-L454W. What is the reason for the analysis change?
=>We thank the reviewer for the observation, we have added the equivalent staining and a new picture in supplemental material to differentiate precursors from differentiated cells.
Results are expressed qualitative analysis, a quantification of the frequency of differences in MeV-infected subpopulations would be important to support the results.
=>We have repeated multiple times the experiments to get better overview of the frequencies which are now in the manuscript as a new figure.
The authors then carried out the quantitative analysis by flow cytometry of the infection of the different cell subpopulations. For wt 105 PFU were used while for L454W 104. Can the analysis be performed with the same PFU for both strains? Could this difference have an impact on cellular tropism?
=>We apology. We in fact used 2.5x104 PFU of L454W virus and 6x104 PFU of wt virus. We have corrected and homogenized in the manuscript. Unfortunately, using 2.5x 104 PFU of wt does not allow getting enough infection to visualize positive cells by flow cytometry (or too few to avoid bias) alternatively putting more than 2.5x 104 PFU of L454W variant rapidly leads to syncytia formation completely biasing the analysis. We have added a sentence in the text to justify this choice in paragraph 3.2.
For Figures 6 and 7, as for Figures 2 and 3, it would be preferable to combine them in order to facilitate the analysis. In addition, the frequencies of infected cells have been indicated in Figure 6. What is the methodology used for this quantification? How many times has this experience been carried out.
=> Calculation were made by scanning the whole slices to find both positive staining out of n≥4 animals. We obtain the percentage of stained cells among total EGFP positive cells. The main risk is that sometimes positive cells in the most inner part of the slice may be less stained.
For Figure 7, frequencies are missing.
Figure 6 and 7 have been pooled and a new figure with frequencies has been added (figure 4).
The analysis of the impact of treatment with type I interferons on the infection of the different subpopulations carried out in Fig. 8 requires further investigation to substantiate the presented results.
In fact, the analysis by flow cytometry seems to be carried out only after treatment with type I interferon. However, the results presented in figure 4 already show a "low" level of infection for the astrocytes and the microglial cells. The comparison with and without treatment with type I interferon is therefore important to show the decrease in the infection of these two cellular subpopulations and to confirm their antiviral status upon IFN-1 response.
=>We thank the reviewer for the suggestion. We have completed the analyses with confocal microscopy and added frequencies in Figure 4.
Moreover, for Figure 8B, the total percentage of cells is close to 170/180% illustrating the problem of double/triple positivity of cells which could impact even slightly the proportions of the different cell subpopulations.
=>We agree with reviewer’s observation. As mentioned to reviewer 1, we tried to get a maximum of information by integrating low positive cells. The side effect of that is that we both have double positive cells (mainly Neurons/oligodendrocytes) but also some of the cells with more autofluorescence. We have thus increased the stringency of our threshold to come back closer to 100%, taking this time the risk of loosing knowing that microscopy analysis would complete the observation.
Minor points
Fig 2, 3 and S1, it would be preferable to use identical magnifications as well as the same order of representation
=>We apology. We tried to play with magnification to show better colocalization or not. Frequencies are now represented as a separated figure to complete observations.
Legend for figure 4: inversion between A and B
=>We have modified in the new figure 3 legend.
Fig 6 and 7 Tuj is shown instead of tuj-1
=>we have corrected the figures.
Fig 8A, global quantification of infected cells would provide quantitative support for the results
=>We have added the values in the paragraph 3.5.

Round 2
Reviewer 1 Report
The manuscript has been revised well. I think this manuscript will be acceptable.